# Evaluation of Particleboards Made from Giant Reed (*Arundo donax* L.) Bonded with Cement and Potato Starch

**DOI:** 10.3390/polym14010111

**Published:** 2021-12-29

**Authors:** Aranzazu Alejandra Ferrandez-García, Teresa Garcia Ortuño, Manuel Ferrandez-Villena, Antonio Ferrandez-Garcia, Maria Teresa Ferrandez-García

**Affiliations:** Department of Engineering, Universidad Miguel Hernández, 03300 Orihuela, Spain; aranfer2@gmail.com (A.A.F.-G.); tgarcia@umh.es (T.G.O.); antonio.ferrandezg@umh.es (A.F.-G.); mt.ferrandez@umh.es (M.T.F.-G.)

**Keywords:** composite, MOR, MOE, IB, panel

## Abstract

There is a general concern about the rationalization of resources and the management of waste. Plant residues can contribute to the development of new non-polluting construction materials. The objective of this study was to valorize a plant residue such as the giant reed and obtain a particleboard with cement using potato starch as a plasticizer in a manufacturing process involving compression and heat. The influence of cement and starch in different proportions and its stability over time were analyzed. Finally, their physical and mechanical properties were evaluated and compared to European Standards. High-quality sustainable particleboards (boards with high structural performance) were obtained and can be classified as P6 according to European Standards. Mechanical properties were improved by increasing the starch content and pressing time, whereas greater resistance to water was obtained by increasing the cement content. Giant reed particles seem to tolerate the alkalinity of the cement since there was no sign of degradation of its fibers. The use of these residues in the manufacture of construction materials offers a very attractive alternative in terms of price, technology and sustainability.

## 1. Introduction

Environmental awareness is increasing in society and with it, public concern about the rationalization of resources and the management of waste. One way to collaborate in solving both problems is by increasing the use of construction materials composed of plant fibers, given these products are easily recyclable and are not aggressive with the environment. The recovery of this waste is also in agreement with European policies related to the environment [1,2,3]. The manufacturing of products from plant residues avoids their elimination through incineration, reducing greenhouse gas emissions by fixing carbon during their life cycle. This could help EU member states fulfill their agreements ensuring that accounted greenhouse gas emissions from land use, land use change or forestry (LULUCF) are balanced by at least an equivalent accounted removal of CO_2_ from the atmosphere in the period 2021 to 2030 [4].

Cement-bonded wood composite panels are not a novel concept, having been on the market during the past century [5]. Cement traditionally has been used to strengthen wood composites improving their mechanical performance [6,7], fire retardance, water resistance and insulation [8]. These products are currently used by the construction industry in applications such as walls, roof sheathing and tiles, floor, fences and sound barriers [9,10,11]. However, due to a decreasing availability of wood, there has been a deterioration in the mechanical properties of these commercial composites.

Studies are currently oriented towards the production of new generation composite boards with lignocellulosic residues of agricultural origin. The development of bio-renewable materials mixed with cement provides added value for the agricultural waste market. The main advantages of using lignocellulosic materials as reinforcement in cement are their low density, low cost, biodegradability, availability of large varieties of fibers throughout the world and the promotion of a new agricultural economy [12,13].

A variety of investigations of plant fibers with cements have been studied: rattan [14], bamboo [15], rice husk [16], sisal [17,18,19], coconut [20,21,22,23,24,25,26], sugar cane bagasse [27,28,29], babaçu [30], banana [31,32], coconut and abaca [33], oil palm [34,35,36], canary palm [37], arhar [38], agave lecheguilla [39], date palm [40], hemp [41,42,43,44,45], kenaf [46], giant reed [47], hazelnut shell, wood and tea [48], cork [49], jute [50,51] and corn [52].

Composites made of natural fibers absorb a large amount of water causing cracking due to the swelling of the fibers. The initial curing process of cement compounds with plant biomass is problematic due to the loss of water absorbed by the fibers. However, later internal curing is favored by the release of part of the water that the biomass had captured. Natural fibers with cement composites are more susceptible to a lack, rather than an excess, of water. High amounts of water in the cement–fiber mix cause the segregation of the mixture, whereas small amounts of water make compaction of the mixture difficult and favors the presence of heterogeneities.

Another issue is that plant fibers suffer degradation when in contact with cement. Some investigations have focused on the modification of the surface of the fibers in order to prevent degradation [40,46,53,54,55,56]. The properties of these composites are limited by a combination of factors such as heterogeneity, wettability and chemical compatibility of the natural fibers with cement. Composites reinforced with lignocellulosic material present a great variability in their mechanical properties in cement boards due to the deterioration of their properties because of alkaline degradation (hydrolysis) and fiber mineralization [57]. These mechanisms produce changes in the chemical composition of the fibers that cause a reduction in strength and degradation of the polymeric matrix, of the fiber/polymer matrix interfacial bond [57] and also of plant fibers resulting in a delay in the cement hydration [31].

Various components of the biomass, such as soluble sugars or low molar mass hemicelluloses, have adverse effects on the preparation and performance of concrete [58]. Therefore, selecting biomass sources with a low content of these compounds would minimize these drawbacks. Research on starch-based lightweight concrete has investigated the effect of its polysaccharides on cement [59,60], the retardant properties of the starch for cement setting [61], the dispersion mechanism of sulfonated starch as a water-reducing agent for cement [62], jute–cement panels with different proportions of starch [63] and particles of palm tree with cement and starch [37].

Different manufacture parameters of panels made of cement with lignocellulosic materials have been analyzed, concluding that cemented panels are high-strength construction materials. However, their industrial production requires a high investment and further research is needed to reduce cost manufacturing [64] and to evaluate the effects of fiber pretreatment methods and alternative curing methods for the long-term performance of these composites [12,65].

Giant reed (*Arundo donax* L.) is one of the largest species of grass growing in Mediterranean regions. It is a wild perennial plant to which no genetic improvement or genotype selection has been made. It grows annually, with average heights of 4 m and a mean thickness of 4 cm. Reed has traditionally been used in many Mediterranean countries. In the southeast of Spain, it was used in construction as part of the roof and floor up to the beginning of the 20th century. However, it is now in disuse and has become an environmental problem since it forms dense reed beds along river banks. When the water level rises, the reeds are uprooted and carried away on the current, forming large masses that block watercourses, causing flooding and sweeping away any structure that gets in their way. In the Segura River in Spain, the authorities are forced to make significant economic investments in order to keep reeds controlled, hence they are cleaned and processed in authorized landfills [66].

Several authors used giant reed biomass to obtain particle-boards with different adhesives. A study on multilayer panels of oriented particles [67] found that these boards have good mechanical behavior and were therefore suitable for use in load-bearing structures according to European Standards [68]. With urea formaldehyde (UF), Garcia Ortuño et al. [69] obtained panels with good properties of modulus of rupture (17.67 N/mm^2^), modulus of elasticity (3025.90 N/mm^2^) and internal bounding strength (1.31 N/mm^2^) that could be commercialized. Giant reed particleboards have also been manufactured with unmodified starches [70] and presented good mechanical properties (modulus of rupture of 16.20 N/mm^2^, modulus of elasticity of 2520.97 N/mm^2^ and internal bounding strength of 0.39 N/mm^2^) through a cyclical process of humidification, heat and pressure. The best properties were found when potato starch was used as an adhesive. In other tests of giant reed with cement mortars, Shon [71] observed that the thermal conductivity and density of the concrete decreased.

Plant residues can contribute to the developing of new non-polluting construction materials. The objective of this research was to valorize a plant residue such as the giant reed and obtain a particleboard, adding a very small amount of cement in comparison with the wood–cement composites and using potato starch as a plasticizer in a manufacturing process involving compression and heat. The influence of cement and starch in different proportions and its stability over time were analyzed. Finally, the particleboard’s physical and mechanical properties were evaluated and compared to European Standards [68] in order to verify whether it could be used as a building material.

## 2. Materials and Methods

### 2.1. Materials

The materials used were residues of giant reed (*Arundo donax* L.) and different proportions of CEM II/B-LL 32.5 N Portland cement, potato starch (*Solanum tuberosum*) and water.

The giant reed biomass (Figure 1) was obtained from clearing the banks of the Segura River in southeast Spain. The reeds were laid out in a vertical position to dry outdoors for 12 months until their relative humidity was 8.2 ± 0.4%. They were then cut and shredded in a blade mill. The particles were collected in a vibrating sieve and only those that passed through the 0.25 mm sieve were selected.

Potato starch from the food industry, with 90% purity, was used as a plasticizer. Chemically, starch is a mixture of two similar polysaccharides: amylose and amylopectin. Potato starch typically contains large oval granules and gels at a temperature of 58–65 °C. Water was taken directly from the mains water supply, with an average temperature of 20 °C.

### 2.2. Manufacturing Process

The manufacturing process consisted of combining dry reed particles with cement in different proportions in weight (0, 10 and 20%) and starch (0, 5 and 10%). Then, 10% water was sprayed onto the mixture before stirring it for 15 min in a blender (LGB100, Imal s.r.l., Modena, Italy) until homogenized.

The mat, formed in a mold of dimensions 600 mm × 400 mm, was then placed in a hot press with a force of 2.6 MPa, a temperature of 100 ºC and four different times (1, 2, 3 and 4 h). Subsequently, the boards were cooled to room temperature. A total of 132 panels were made, comprising seven types with four different classes (the 28 different configurations are shown in Table 1 and Table 2).

Twenty-eight days after manufacture, four specimens of each type (A to G) and class (1 to 4) were cut to the appropriate dimensions as indicated in European Standards [72] in order to carry out the tests needed to characterize the mechanical, physical and thermal properties of each of the boards being studied (Figure 2). Three hundred and sixty-five days after manufacture, four boards of class 2 from types A to E were cut and also tested.

### 2.3. Methods

The method followed was experimental. The tests were conducted in the Materials Strength Laboratory of the Higher Technical College of Orihuela at Universidad Miguel Hernández, Elche. The values were determined according to European Standards established for wood particleboards [73].

After they were manufactured and cut, density [74], thickness swelling (TS) and water absorption (WA) after 2 and 24 h immersed in water [75], internal bonding strength (IB) [76], modulus of elasticity (MOE) and modulus of rupture (MOR) [77] were measured (Table 3). Later, the boards were evaluated according to European Standards [68]. In order to assess the resistance of the reed particles to the alkalinity of the cement, MOR and MOE tests were performed after 365 days on four class 2 panels of types A to E.

The morphology of the inside of the experimental panels was examined using a scanning electron microscope (SEM) (Hitachi model S3000N, Hitachi, Ltd., Tokyo, Japan) equipped with an X-ray detector (Bruker XFlash 3001, Billerica, MA, USA). For the observations, images of fractured 5 mm × 5 mm cross-sections of the panels were taken.

The moisture content of the material was measured with a laboratory moisture meter (model UM2000, Imal s.r.l., Modena, Italy). The water immersion test for the panels was carried out in a heated tank (Model 76-B0066/B Water Bath, Equipos de Ensayo Controls S.A., Toledo, Spain).

The mechanical tests and density were performed with the universal testing machine (model IB700, Imal s.r.l., Modena, Italy), which complies with the velocity of 5 mm·min^−1^ for the bending test and 2 mm·min^−1^ for internal bonding strength.

For the statistical analysis, SPSS v. 26.0 software (IBM, Chicago, IL, USA) was used. Analysis of variance (ANOVA) was performed for a significance level of α < 0.05. The standard deviation was obtained for the mean values of the tests.

## 3. Results

### 3.1. Physical Properties

Average density is showed in Figure 3, with all boards considered to have medium-high density. Boards made with starch alone (F, G) had a density range from 866 to 988 kg/m^3^, those with cement alone (D, E) were from 942 to 1026 kg/m^3^ and those with starch and cement (A, B, C) were from 990 to 1116 kg/m^3^. Boards manufactured with a mix of cement and starch have higher density than using these two components separately. The boards with the highest density were B4 class, with a composition of 5% potato starch and 20% cement. They were in the hot plate press for 4 h, reaching an average density of 1116 kg/m^3^.

Figure 4 shows the mean values of the thickness swelling tests (TS) and water absorption (WA) after 24 h of immersion in water. The boards with starch alone (F, G) had a TS from 37.2 to 56.3%, with cement (D, E) ranging from 26.7 to 58.5% and starch and cement (A, B, C) from 11.5 to 23.8%. WA follows the same tendency of TS. Type B, made with 5% starch and 20% cement, had the lowest WA value between 37.27 and 43.85%. Panels made of cement and starch achieved a great result and could be classified as structural boards.

### 3.2. Mechanical Properties

Modulus of rupture (MOR) and modulus of elasticity (MOE) results follow a similar pattern, as shown in Figure 5. MOR performance of the boards made with starch (F, G) ranged from 7.89 to 18.53 N/mm^2^, with cement (D, E) ranging from 6.21 to 18.04 N/mm^2^ and starch and cement (A, B, C) from 12.59 to 27.26 N/mm^2^. MOE values with starch (F, G) ranged between 1663.3 and 2763.7 N/mm^2^, with cement (D, E) ranging from 1234.8 to 3080.1 N/mm^2^ and starch and cement (A, B, C) from 1846.1 to 4287.4 N/mm^2^.

Type A and C boards, using starch and cement and a press time of 1 h, can be classified as P2. After being in the press for 4 h, these panels achieved the P7 requirements. The mechanical results obtained were very high, showing the good mechanical properties that can be achieved with boards made of giant reed–cement–starch.

The average values obtained for internal bonding strength (IB) (Figure 6) varied between 0.20 and 0.28 N/mm^2^_,_ with starch (F, G) with cement (D, E) ranging between 0.15 and 0.36 N/mm^2^ and starch and cement (A, B, C) between 0.36 and 0.89 N/mm^2^.

The results of the test indicate that the contribution of starch improved the properties of the boards, as was observed in the palm–cement boards [37]. Board types A, B and C could be classified as P7, whereas if only cement or starch was used, the boards only met the P1 requirements.

### 3.3. Assessment of the Durability of the Fibers

To verify that fibers were not degraded in contact with cement, four extra boards of classes A2, B2, C2, D2 and E2 (the types containing cement) were made. These extra specimens were cut for the mechanical tests that were carried out 365 days after manufacture and compared to the others specimens from the same class. The results obtained from MOR and MOE are shown in Figure 7. It can be concluded that boards made with starch and cement (classes A2, B2 and C2) had a significant increase in the MOR and MOE values at 365 days of setting. The boards that only contained cement (classes D2 and E2) presented similar values.

This demonstrates that giant reed particles are not degraded by the alkalinity of the cement, as no deterioration was observed in their mechanical properties. Boards made of reed with cement and starch showed a notable improvement over time that may be justified by the beneficial effect that starch could exert on the setting of the cement. It is possible that the water absorbed by the starch at the beginning of the manufacturing process was gradually transferred to the cement, favoring its subsequent hydration.

### 3.4. Statistical Analysis

The ANOVA shown in Table 4 indicated that density depends on the type of board, whereas the time in the press has no influence. Density also depends on the amount of starch used but not the cement. TS and WA are dependent on the type of board and the amount of cement and starch in the mix. Pressing time is not influential in the physical properties. MOR, MOE and IB are influenced by all factors: type of board, pressing time and amount of cement and starch added to the mix.

### 3.5. SEM Observations

Scanning electron microscope (SEM) observations were made to investigate the interaction between the three components of the experimental boards. Using a standard method [78], the mineralogical evolution of cement during hydration of the boards was analyzed. Micrographs were obtained at 8 and 28 days after the manufacture of the boards of class 2, which contained 20% cement and 10% starch and had been pressed for 2 h. Samples for SEM were polished and plated with gold.

Figure 8a shows that starch has gelled and enveloped the cement grains that were being hydrated. Figure 8b shows how cement grains were hydrating eight days after panels were made. It can be differentiated that alite (calcium silicate) and belite (dicalcium silicate) were hydrating, forming tobermorite grains. Hydration cracks were also found.

Figure 9a shows a cube of gypsum (calcium sulfate) that was added to the clinker to decrease the solubility of the aluminate. This cube is surrounded by crystals of tricalcium silicate (C_3_S) transformed into tobermorite gel (hydrated calcium silicates), whose grains began to coalesce. In Figure 9b, tobermorite is seen over an organic gel, indicating that the starch has gelled.

Figure 10a shows that tobermorite gel has bonded, forming a continuous matrix, while in Figure 10b some tobermorite grains have coalesced forming a continuous layered matrix. Some thin hexagonal plates can be seen, indicating the presence of portlandite [Ca(OH)_2_].

Figure 11a shows the halo reaction of belite (dicalcium silicate). In Figure 11b, alite crystals (calcium silicate) in the hydration process are seen, forming tobermorite grains which were forming a gel.

Portlandite (calcium hydroxide) over tobermorite (hydrated calcium silicates) is observed in Figure 12a, with no ettringite found. Figure 12b shows giant reed fibers glued together, with no degradation due to the alkalinity of the cement observed.

There is no visible sign of microcracks on the surface of the boards 8 or 28 days after setting in Figure 13. Even though other authors observed degradation of plant fibers in contact with cement by alkaline hydrolysis, the hydration process of the experimental boards of this study was optimal.

To elucidate the mechanism involved in why boards with starch (types A, B and C) have better properties than boards without starch (D and E), a micrograph of board type E2 (no starch, 20% cement and 2 h in the press) after 28 days of setting is shown in Figure 14.

It shows cement components without hydration. This indicates that there was a water shortage that prevented a correct absorption of water by the cement. It is possible that water contained in the mixture before hot pressing was attracted by the reed particles and then evaporated when pressed. Boards made with starch could have retained this water and later released it to the cement to hydrate it properly, thereby improving the mechanical performance of the board.

## 4. Discussion

Particleboards are classified based on their mechanical properties and dimensional stability by European Standards [68], which establishes minimum requirements of their properties. Their grades range from P1 to P7 (Table 5).

According to Table 6, boards whose only binder was 20% cement (type E) had lower mechanical properties, which indicates that hydration of the cement was not adequate and the water initially contained in the mixture was absorbed by the giant reed particles. When pressure and heat were applied to these boards, water evaporated, preventing the correct setting of the cement.

The boards with the best performance were types A4 and C4, which met European Standards [68] and could be classified as P6 (high performance structural load-bearing particleboard to use in dry conditions). In general, boards with the best properties are those in which a mixture of cement and starch has been used as a binder.

Higher proportions of cement produced lower TH, whereas higher proportions of starch resulted in better mechanical performances. Starch could have retained part of the water previously added onto the mixture to later transfer it to the cement, favoring its correct hydration. This confirmed a study that suggested that in order to improve the properties of cementitious compounds, an additive of cellulose and starch was added because it behaves as a good water-retention agent [62].

The two manufacturing variables considered in this study were the type of binder and the time in the hot plate press. The type of binder is the most influential parameter since all physical and mechanical properties varied according to it (density, TS, WA MOR, MOE and IB). The pressing time affected half of the properties tested (MOR, MOE and IB).

Several authors noticed that during the setting time of cement composites with plant fibers, there was a delay in the formation of hydration products [15,28,79]. They attributed it to different fiber components, especially sugars. A different study indicated that pectins contained in the jute fibers combined with the cement functioned as a growth inhibitor of calcium silicate hydrate (CSH) [41].

SEM micrographs showed calcium compounds at eight days of setting and therefore the hydration process had not been delayed. In addition, micrographs showed how the gelled starch enveloped the cement grains, which would make difficult its dilution and the interaction of the reed components with the cement.

The hydration of the cement in the experimental boards is in accordance with the study carried out with cement without plant fibers at seven days [78]. Some research [57] affirms that plant fibers degrade in contact with cement by alkaline hydrolysis. However, in this work, no signs of degradation of giant reed fibers were observed at 28 days. Another investigation [31] indicated that composites made of cement and plant fibers decrease performance in MOR and MOE after one year of setting due to carbonation of the matrix followed by lixiviation and progressive microcracking. In this paper, results appear to differ, with the MOR and MOE behavior of the panels even increasing, indicating that there was no incompatibility between the three components (giant reed, cement and starch).

Other researchers obtained composites made of cement and wood or plant fibers. Their results are shown in Table 7. Pine wood–cement composites have lower performance than the boards of this study. It is possible that this type of wood requires large amounts of cement to comply with European Standards [68]. With eucalypt and rubberwood or with pretreated coconut coir, the results of TS are better than in this study. Nonetheless, one of the uses of these composites is in humid conditions such as terraces or façades. It is possible that the pretreatment of the coconut or the amount of cement could improve the TS of the panels of this study. The behavior of canary palm composites with amounts of cement similar to this investigation is sufficient to be commercialized as P2 [68]. However, their properties are behind the giant reed–cement composites of this study.

Industrial boards are normally manufactured with a wood/cement ratio of 1/5.5. The boards in this work have been made with a small amount of cement, with the ratio in weight of giant reed/cement being 1/0.2 maximum. Pressing time was a maximum of 4 h at 100 °C, while industrial boards need 8 h at 75 °C and drying times of 8 h at 90 °C. The experimental boards described in this study have been made with less time and with a much lower proportion of cement than in the production of industrial wood–cement boards. This represents significant energy savings in their manufacture and makes them potentially a more ecological product.

## 5. Conclusions

The effects of the binder (starch–cement) on the physical and mechanical properties of boards made of giant reed (*Arundo donax* L.) particles were investigated.

High-quality sustainable particleboards (boards with high structural performance) were obtained using giant reed and can be classified as P6 according to European Standards. Mechanical properties were improved by increasing the starch content, whereas greater resistance to water was obtained by increasing the cement content.

The pressing time of the boards in the hot plate press at 100 °C had a great influence in MOR, MOE and IB, which increased performance. Further tests are needed to determine the optimum pressing temperature and time for the manufacturing process.

Cement is mineralized in the giant reed–cement–starch particleboards following the same hydration process that occurred without fibers or additives. Giant reed particles seem to tolerate the alkalinity of the cement since there was no sign of degradation of its fibers, with MOR and MOE even increasing at 365 days of setting.

Using giant reed particles, cementitious composites can be made which could be used in the manufacture of various construction components. Therefore, the use of these residues in the manufacture of construction materials offers a very attractive alternative in terms of price, technology and sustainability.

## Figures and Tables

**Figure 1 polymers-14-00111-f001:**
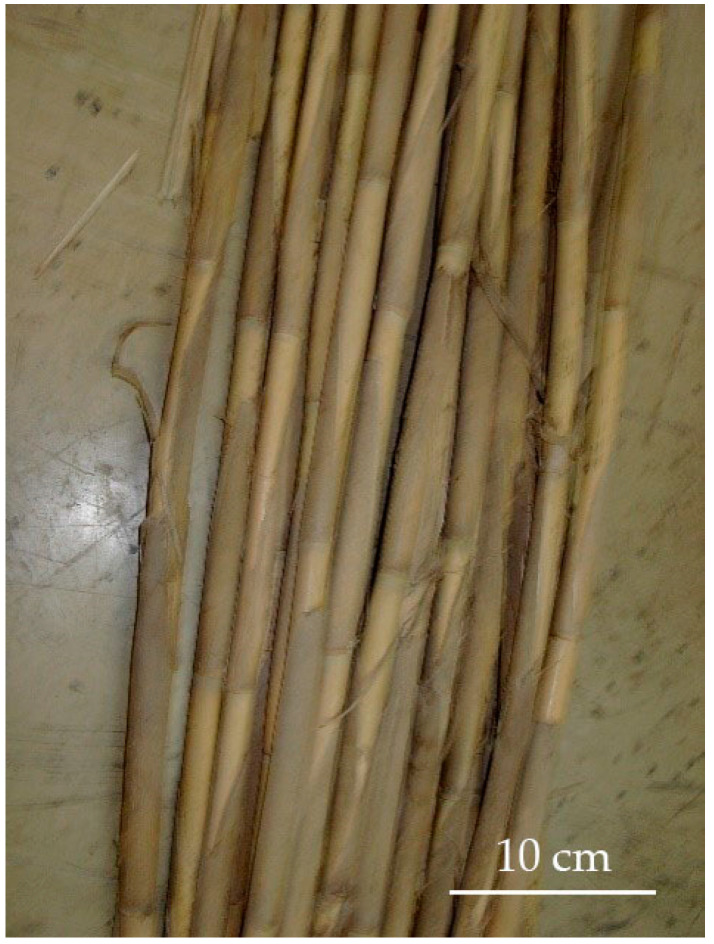
Giant reed used in the manufacturing of the panels.

**Figure 2 polymers-14-00111-f002:**
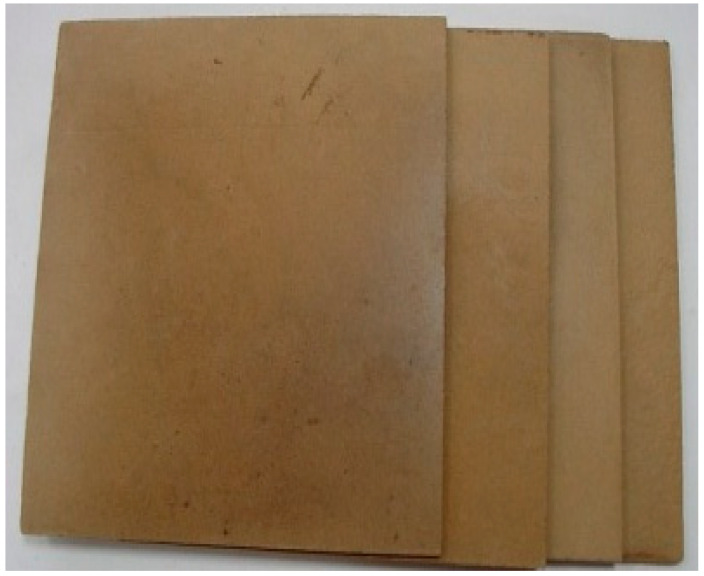
Giant reed–cement–starch panels.

**Figure 3 polymers-14-00111-f003:**
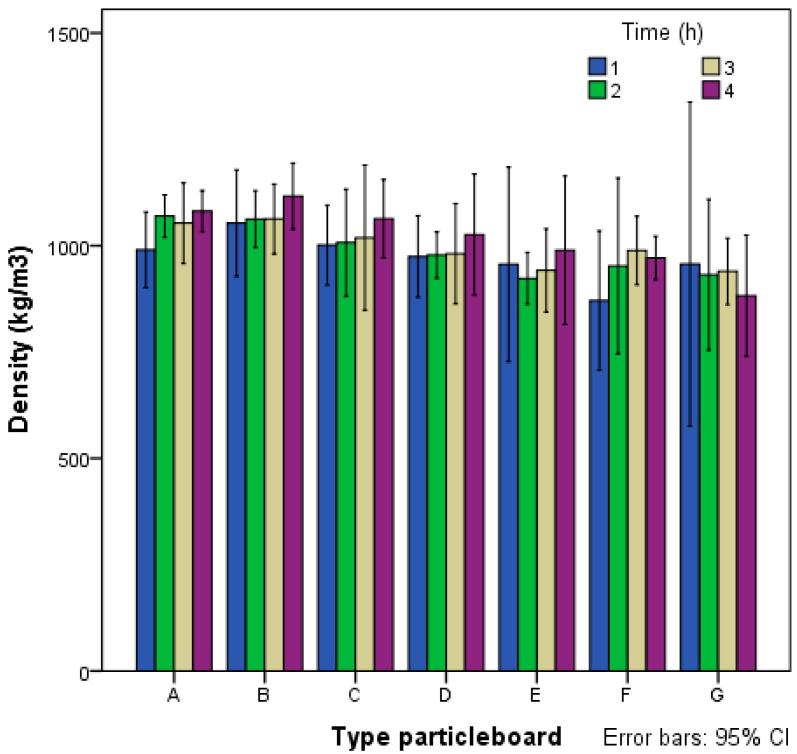
Mean density of type A–G particleboards.

**Figure 4 polymers-14-00111-f004:**
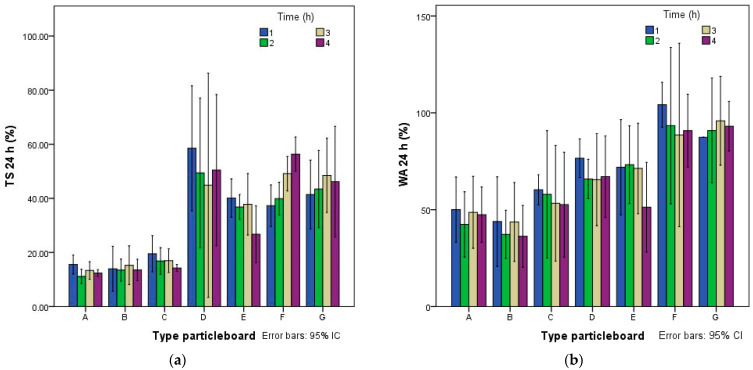
(**a**) Thickness swelling (TS) after 24 h and (**b**) water absorption (WA) after 24 h.

**Figure 5 polymers-14-00111-f005:**
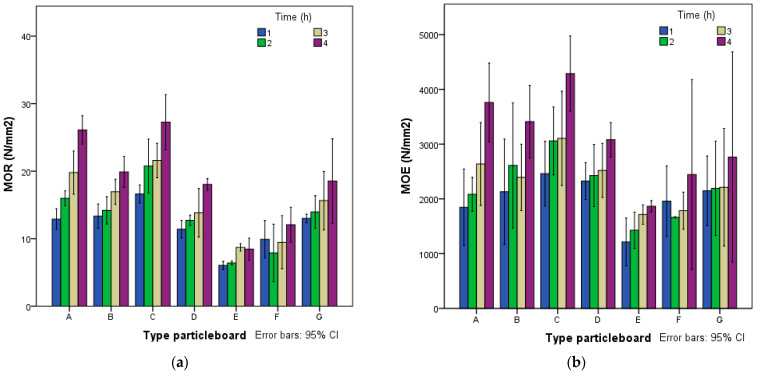
(**a**) Modulus of rupture (MOR) and (**b**) and modulus of elasticity (MOE).

**Figure 6 polymers-14-00111-f006:**
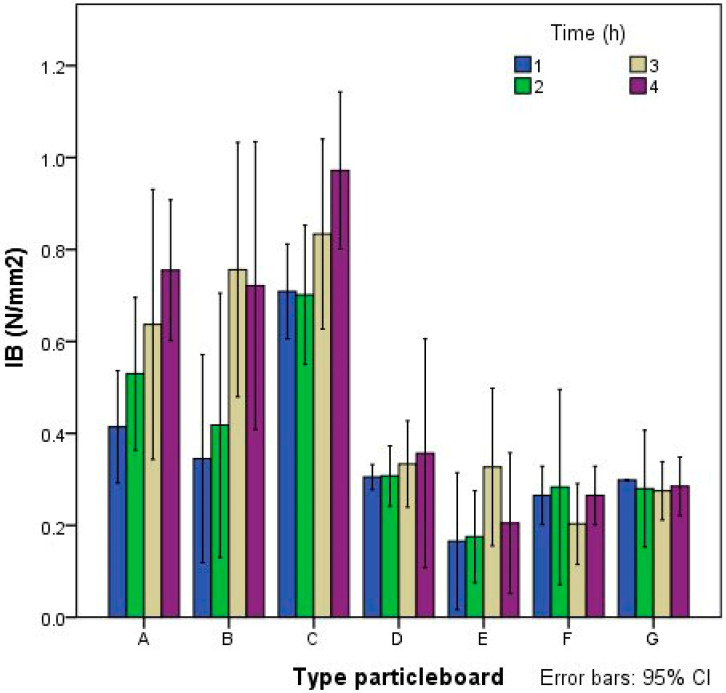
Internal bounding strength.

**Figure 7 polymers-14-00111-f007:**
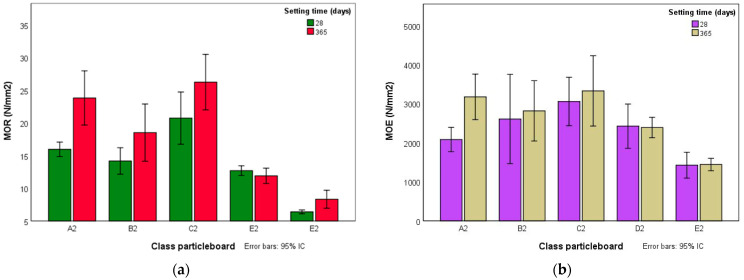
(**a**) Modulus of rupture (MOR) and (**b**) modulus of elasticity (MOE) after 28 and 365 days of cement hydration.

**Figure 8 polymers-14-00111-f008:**
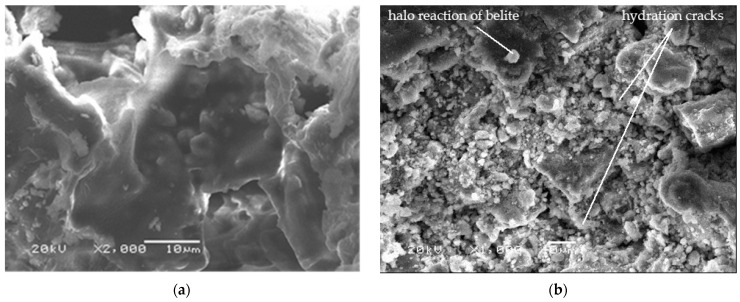
(**a**) Starch gel enveloping cement grains 3 days after hydration and (**b**) cement grains 8 days after hydration.

**Figure 9 polymers-14-00111-f009:**
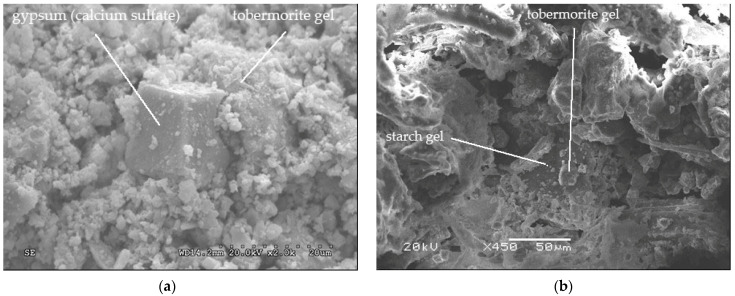
(**a**) Gypsum cube surrounded by tobermorite gel 3 days after hydration and (**b**) tobermorite over an organic gel 8 days after hydration.

**Figure 10 polymers-14-00111-f010:**
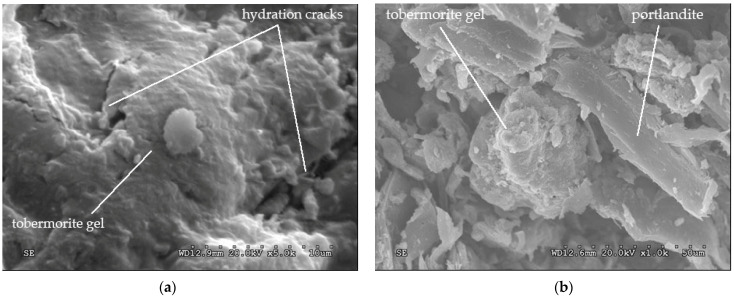
(**a**) Tobermorite gel 8 days after hydration and (**b**) tobermorite and crystalized portlandite 8 days after hydration.

**Figure 11 polymers-14-00111-f011:**
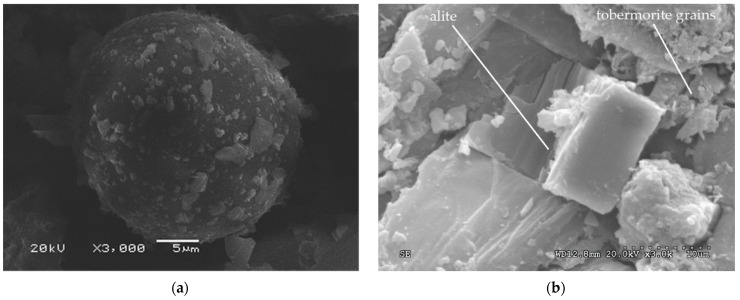
(**a**) Belite 8 days after hydration and (**b**) tobermorite, alite and portlandite 8 days after hydration.

**Figure 12 polymers-14-00111-f012:**
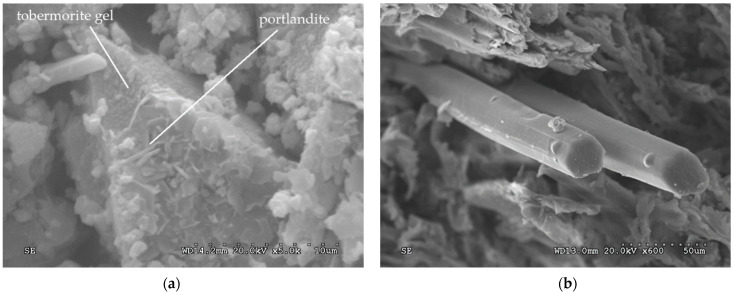
(**a**) Portlandite over tobermorite 8 days after hydration and (**b**) giant reed fibers 8 days after hydration.

**Figure 13 polymers-14-00111-f013:**
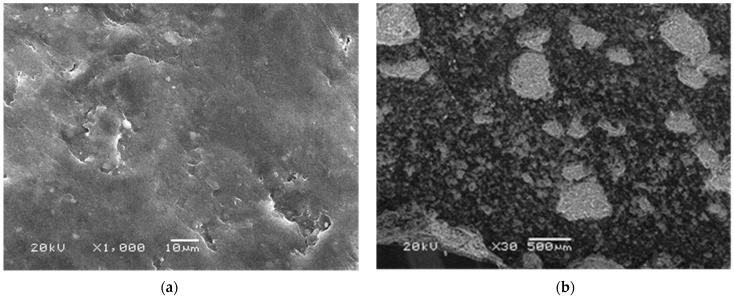
Surface of the board (**a**) 8 days after hydration and (**b**) 28 days after hydration.

**Figure 14 polymers-14-00111-f014:**
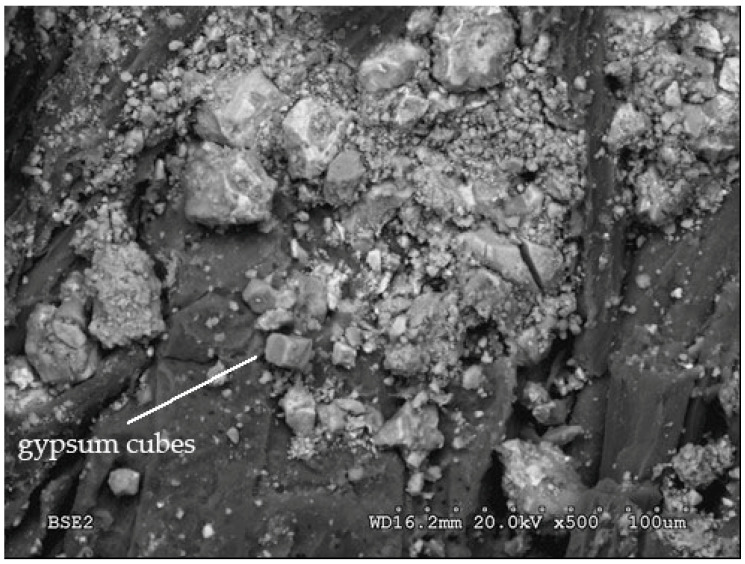
Cross-sectional micrograph of a board type E2 28 days after hydration.

**Table 1 polymers-14-00111-t001:** Types of panels manufactured.

Type	Number of Panels	Weight Dosage (%)	
		Starch	Cement
A	20	10	20
B	20	5	20
C	20	10	10
D	20	-	10
E	20	-	20
F	16	5	-
G	16	10	-

**Table 2 polymers-14-00111-t002:** Division of panels in classes according to their pressing time.

Pressing Time (h)	Class	Number of Panels
1	1	4
2	2	8 ^1^
3	3	4
4	4	4

^1^ Only for types A to E. For types F and G, the number of panels was four.

**Table 3 polymers-14-00111-t003:** Characteristics of the tests performed.

Test	N of Replicates (Per Panel)	N of Replicates (Total)	Size of the Specimens	Equipment Used
Relative Humidity	3	396	20 g	Model UM2000, Imal s.r.l.
Density	6	792	50 mm × 50 mm	Model IB700, Imal s.r.l.
Thickness Swelling (TS)	3	396	50 mm × 50 mm	Model 76-B0066/B Water Bath, Controls S.A. Model UM2000, Imal s.r.l.
Water Absorption (WA)	3	396	50 mm × 50 mm	Model 76-B0066/B Water Bath, Controls S.A. Model UM2000, Imal s.r.l.
Modulus of Rupture (MOR)	6	792	150 mm × 50 mm	Model UM2000, Imal s.r.l.
Modulus of Elasticity (MOE)	6	792	150 mm × 50 mm	Model UM2000, Imal s.r.l.
Internal Bonding Strength (IB)	3	396	50 mm × 50 mm	Model UM2000, Imal s.r.l.

**Table 4 polymers-14-00111-t004:** ANOVA of the test results.

Factor	Properties	Sum of Squares	d.f.	Half Quadratic	F	Sig
Type of panel	Density (kg/m^3^)	246,938.733	6	41,156.455	8.851	0.000
	TS 24 h (%)	24,117.087	6	4019.514	79.859	0.000
	WA 24 h (%)	30,972.016	6	5162.003	24.673	0.000
	MOR (N/mm^2^)	2203.595	6	367.266	22.757	0.000
	MOE (N/mm^2^)	2.572 × 10^7^	6	4.286 × 10^6^	8290	0.000
	IB (N/mm^2^)	3.947	6	0.658	10.880	0.000
Pressing time	Density (kg/m^3^)	44,648.253	3	14,882.751	2.285	0.083
	TS 24 h (%)	421.488	3	140.496	0.496	0.686
	WA 24 h (%)	1060.883	3	353.628	0.708	0.549
	MOR (N/mm^2^)	1081.134	3	360.378	13.461	0.000
	MOE (N/mm^2^)	2.348 × 10^7^	3	7.827 × 10^6^	14.943	0.000
	IB (N/mm^2^)	0.995	3	0.332	3.771	0.013
Starch (%)	Density (kg/m^3^)	225,752.148	2	112,876.074	24.920	0.000
	TS 24 h (%)	22,366.496	2	11,183.248	170.680	0.000
	WA 24 h (%)	28,956.715	2	14,478.357	71.060	0.000
	MOR (N/mm^2^)	1686.149	2	843.074	40.973	0.000
	MOE (N/mm^2^)	14,861,771.141	2	7,430,885.571	12.256	0.000
	IB (N/mm^2^)	3.682	2	1.841	50.932	0.000
Cement (%)	Density (kg/m^3^)	7589.261	1	7589.261	1.391	0.243
	TS 24 h (%)	80.590	1	80.590	7.165	0.010
	WA 24 h (%)	1418.878	1	1418.878	5.653	0.021
	MOR (N/mm^2^)	307.374	1	307.374	15.118	0.000
	MOE (N/mm^2^)	8,513,888.803	1	8,513,888.803	12.613	0.001
	IB (N/mm^2^)	0.773	1	0.773	16.647	0.000

d.f.: degrees of freedom. F: Fisher–Snedecor distribution. Sig: significance.

**Table 5 polymers-14-00111-t005:** Classification of experimental panels based on European Standards [68].

Grade	Definition
P1	Boards for general use in dry conditions
P2	Boards for indoor application (including furniture) in dry conditions
P3	Non-structural boards for use in humid conditions
P4	Structural boards for use in dry conditions
P5	Structural boards for use in humid conditions
P6	High performance structural boards for use in dry conditions
P7	High performance structural boards for use in humid conditions

**Table 6 polymers-14-00111-t006:** Physical and mechanical properties of the experimental panels.

Type of Panel	Classification EN 312 [68]	MOR (N/mm^2^)	MOE (N/mm^2^)	IB (N/mm^2^)	TS 24 h (%)
A1	P2	12.89 (1.44)	1846.04 (368.20)	0.41 (0.13)	15.54 (3.37)
A2	P3	15.98 (1.19)	2083.50 (335.15)	0.53 (0.22)	11.12 (2.85)
A3	P4	19.79 (3.06)	2636.88 (521.43)	0.66 (0.27)	13.33 (3.08)
A4	P6	26.11 (2.28)	3759.76 (377.99)	0.76 (0.21)	12.38 (1.35)
B1	P1	13.33 (1.13)	2130.78 (304.21)	0.34 (0.15)	13.96 (5.22)
B2	P2	14.19 (1.63)	2610.01 (520.34)	0.42 (0.23)	13.48 (3.28)
B3	P4	16.94 (1.49)	2392.63 (488.00)	0.79 (0.21)	15.25 (5.77)
B4	P4	19.88 (0.91)	3076.93 (431.62)	0.74 (0.14)	13.49 (1.59)
C1	P2	12.56 (6.07)	2021.89 (461.01)	0.54 (0.19)	24.22 (8.13)
C2	P3	20.76 (2.51)	3058.56 (389.95)	0.71 (0.09)	16.83 (3.09)
C3	P3	21.60 (1.60)	3104.68 (541.19)	0.84 (0.24)	16.95 (2.75)
C4	P6	27.26 (3.29)	4287.44 (554.31)	0.95 (0.33)	14.04 (1.03)
D1	P1	11.41 (0.81)	2325.87 (211.76)	0.31 (0.01)	58.50 (14.50)
D2	P1	12.72 (0.46)	2426.60 (356.39)	0.31 (0.04)	49.42 (0.17)
D3	P1	13.82 (1.44)	2519.48 (198.45)	0.34 (0.03)	44.83 (16.6)
D4	P1	18.04 (0.35)	3080.12 (126.14)	0.36 (0.10)	50.41 (11.20)
E1	-	6.22 (0.22)	1234.77 (131.71)	0.16 (0.11)	42.24 (1.56)
E2	-	6.40 (0.24)	1426.89 (167.47)	0.16 (0.06)	36.79 (3.71)
E3	-	8.71 (0.34)	1713.74 (11.04)	0.33 (0.10)	37.79 (7.15)
E4	-	9.45 (1.02)	1865.42 (64.53)	0.21 (0.09)	26.73 (6.59)
F1	-	9.90 (0.30)	1958.16 (71.76)	0.27 (0.01)	37.28 (1.85)
F2	-	7.90 (0.47)	1663.29 (1.38)	0.28 (0.02)	39.89 (0.67)
F3	-	9.46 (0.43)	1784.03 (37.78)	0.21 (0.01)	49.11 (0.71)
F4	-	12.06 (0.29)	2443.87 (51.51)	0.27 (0.01)	56.34 (0.71)
G1	P1	13.00 (0.07)	2148.25 (70.71)	0.30 (0.01)	41.41 (1.41)
G2	P1	14.04 (0.27)	2192.85 (95.95)	0.28 (0.01)	43.44 (1.59)
G3	P1	15.63 (0.48)	2211.76 (119.50)	0.28 (0.01)	48.49 (1.52)
G4	P1	18.53 (0.69)	2763.68 (72.18)	0.29 (0.01)	46.18 (2.28)
Thickness 6–13 mm [68]	P1	10.50	-	0.28	-
	P2	11.00	1800.00	0.40	-
	P3	15.00	2050.00	0.45	17.00
Thickness 6–10 mm [68]	P4	16.00	2300.00	0.40	16.00
	P5	18.00	2550.00	0.45	13.00
	P6	20.00	3150.00	0.60	16.00
	P7	22.00	3350.00	0.75	10.00

() standard deviation.

**Table 7 polymers-14-00111-t007:** Properties obtained with cement composites.

Source	Material	Ratio Cement/Material	MOR (N/mm^2^)	MOE (N/mm^2^)	IB (N/mm^2^)	TS 24 h (%)
[5]	Pine	3/1	15.8	5495	-	18.5
[9]	50% Eucalypt and 50% rubberwood	8/1	6.40	4090	0.34	1.80
[21]	Coconut coir	2/1	19.94	5315	0.73	3.64
[37]	Canary palm	1/5	15.76	1872	0.68	26.70
This work (C4)	Giant reed	1/10	27.26	4287.44	0.95	14.04

## Data Availability

The data presented in this study are available within the article.

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
