# Peer review of "Evaluation of Particleboards Made from Giant Reed (*Arundo donax* L.) Bonded with Cement and Potato Starch"

_polymers, 2021, doi:10.3390/polym14010111_

Round 1
Reviewer 1 Report
This is an excellent written, presented and discussed paper. Especially the session 3.5 is perfectly written and includes information that is missed from the literature.
I just want to see some more related studies to be included in the introduction session, dealing with the application of cement as a potential binder in wood composites manufacture
Author Response
Point 1: I just want to see some more related studies to be included in the introduction session, dealing with the application of cement as a potential binder in wood composites manufacture.
Response 1: Dear revisor 1, we have included a few references of investigations and applications of wood-cement composites. You can find them in the next paragraph in the manuscript:
Cement-bonded wood composite panels are not a novel concept, having been on the market over the last century [5]. Cement has been used traditionally to strengthen wood composites improving their mechanical performance [6-7], fire retardant, water resistance and insulation [8]. These products are currently used by the construction industry in applications such as walls, roof sheathing and tiles, floor, fences, and sound barriers [7-10]. But due to a decreasing availability of wood, there has been a deterioration in the mechanical properties of these commercial composites.
References:
5. Moslemi, A. A., Pfister, S. C. The influence of cement/wood ratio and cement type on bending strength and dimensional stability of wood-cement composite panels. Wood and fiber science, 1987, 19(2), 165-175.
6. Ferrier, E., Agbossou, A., Michel, L. Mechanical behaviour of ultra-high-performance fibrous-concrete wood panels reinforced by FRP bars. Composites Part B: Engineering, 2014, 60, 663-672. DOI: 10.1016/j.compositesb.2014.01.014
7. Li, M., Khelifa, M., Khennane, A., El Ganaoui, M. Structural response of cement-bonded wood composite panels as permanent formwork. Composite Structures, 2019, 209, 13-22. DOI: 10.1016/j.compstruct.2018.10.079
8. Brahmia, F. Z., Horváth, P. G., Alpár, T. L. Effect of pre-treatments and additives on the improvement of cement wood composite: a review. BioResources, 2020, 15(3), 7288-7308. DOI: 10.15376/biores.15.3.Brahmia
9. Okino, E. Y., De Souza, M. R., Santana, M. A., da S Alves, M. V., de Sousa, M. E., Teixeira, D. E. Cement-bonded wood particleboard with a mixture of eucalypt and rubberwood. Cement and Concrete Composites, 2004, 26(6), 729-734. DOI: 10.1016/S0958-9465(03)00061-1
10. Tittelein, P., Cloutier, A., Bissonnette, B. Design of a low-density wood–cement particleboard for interior wall finish. Cement and Concrete Composites, 2012, 34(2), 218-222. DOI: 10.1016/j.cemconcomp.2011.09.020
11. Fuwape, J. A., Oyagade, A. O. Bending strength and dimensional stability of tropical wood-cement particleboard. Bioresource technology, 1993, 44(1), 77-79. DOI: 10.1016/0960-8524(93)90212-T
Reviewer 2 Report
The objective of this research was to valorize a plant residue such as the giant reed and obtain a particle board with cement using potato starch as a plasticizer in a manufacturing process involving compression and heat. The manuscript is fairly written, significant scientific English review would require, but contain new information and presented nicely. The discussion part contains correct interpretations. It is worth for publishing.
Grammatical errors:
This type of numbers are very difficult to read: 1,663.3. Please do not use comma to indicate the thousands, as you did not do for the density values as well.
Toverify thatfibers...
Theseextra
Belite8 days
porlandite - it should be portlandite!
Interpretation of Fig. 12.b is wrong: if that long particle could be fibers, they should have hole inside, these are solid objects.
MOE and IB,).
The following sentence is still early to conclude, as these boards has not been tested against fire. The meaning of the high cement ratio is required for the fire retardant properties of these type of boards: "The experimental boards described in this study have been made with less time and with a much lower proportion of cement than in the production of industrial wood‐cement boards, which represents significant energy savings in their manufacture and a more ecological product."
Reviewer 3 Report
polymers-1506686
The introduction is well written. The authors provide a general overview of the problem. In order to improve this section, I suggest the authors to provide the following information:
- Current and potential application of the composites. The application is a basis for performance studies.
- Would you please describe the commercial importance of the respective field? Wood cement composites are available on a commercial scale.
- Are the amounts of reed suitable for production on a commercial scale?
L28: Please refine the term »aggressive with the environment« It is hard to understand. Please specify.
L19, which waste? Composite? Reed? Concrete?
L32, authors have to be aware that several EU policies address carbon storage. One of the key ones is LULUCF. The principle is based on the long term balance of organic compounds used in a specific sector. If authors list the carbon storage effect, they must consider international accounting regulations.
L37, are lignocellulosic cement-based composites biodegradable if only part of the composite is degradable?
L56, what are cellulosic composites? Composites based on the cellulose only, or composites reinforced with lignocellulosic material of high cellulose content.
L86, please consider that harvesting of the reed is not allowed in some of the Mediterranean lagoons like Grado, Capodistria and Venetia. Please consider this perspective as well. Similarly, a reed is not allowed to be used (or extremely limited) in some of the central European countries.
L87-96; please provide some values for respective statements. What are good mechanical properties, for example?
L97-103 PLease sharpen the objectives. Can we expect a better performance of reed-cement composites compared to wood-cement. Why?
Materials and methods
L107; please write scientific names in Italic
L111. is the respective drying procedure standard one? How did you determine the MC? How did you mill the reed?
Figure 1; please provide the scale bar within the image. This will give us an impression of the size of the reed columns.
L127-131; did you develop your own procedure for particleboards production, or did you follow the existing one? Please specify.
L149 – 154; please describe respective methods in more detail. Authors have to provide the following information:
- Number of replicates
- Size of the specimens
- Equipment used for testing
This information is needed, as all of the research audience does not have access to EU standards. So this information is a must to understand the testing principle.
L155, which surface did you asses through SEM. How did you prepare the surface? Which vacuum mode, which detector did you use?
L160; why did you use the respective method for MC analysis and not old school oven. Are these methods validated for larger particles I do not have good experience with this method for the samples of more significant dimensions?
Statistics is not addressed.
Results and discussion
Artwork: i recommend the authors to use box and whisker plots, as they are much more informative than columns. It can not be resolved from the graphs what deviation measure stands for.
Authors should perform statistical analysis—only this enables proper comparison of the results.
Statistical analysis should be an integrative part of the results, not separate subchapters. Authors should clearly mark (with letters) which bords belong to the same homogeneity group.
Discussion,
Please compare the results with literature data. What are the values obtained on other cement-based composites? What are the values of the commercial products? At the very moment, the result section looks like a technical report. There is a lack of comparison of the result. There is no perspective?
Please explain the EU classification system, as all of the audience might not be familiar with this system.
I declare no conflicts of interest
Round 2
Reviewer 3 Report
The authors have revised the manuscript, they have put a lot of effort into the revision. The authors have provided detailed responses with clearly marked comments/arguments. Statements are supported by respective literature. In addition, changes are visible from the manuscript as well. As authors have addressed all of the open questions, and as I do not have additional suggestions, I recommend the editors accept the respective manuscript.